# Variants in the Myostatin Gene and Physical Performance Phenotype of Elite Athletes

**DOI:** 10.3390/genes12050757

**Published:** 2021-05-17

**Authors:** Valentina Ginevičienė, Audronė Jakaitienė, Erinija Pranckevičienė, Kazys Milašius, Algirdas Utkus

**Affiliations:** 1Department of Human and Medical Genetics, Institute of Biomedical Science, Faculty of Medicine, Vilnius University, LT-01513 Vilnius, Lithuania; audrone.Jakaitiene@mf.vu.lt (A.J.); erinija.pranckeviciene@mf.vu.lt (E.P.); algirdas.utkus@mf.vu.lt (A.U.); 2Academy of Education, Vytautas Magnus University, LT-44244 Kaunas, Lithuania; kazys.milasius@vdu.lt

**Keywords:** sprint and power, endurance, myostatin, polymorphism, phenotype, genotype

## Abstract

The *MSTN* gene is a negative regulator of muscle growth that is attracting attention as a candidate gene for physical performance traits. We hypothesised that variants of *MSTN* might be associated with the status of elite athlete. We therefore sought to study the potential role of *MSTN* in the physical performance of athletes by analysing the whole coding sequence of the *MSTN* gene in a cohort of Lithuanian elite athletes (*n* = 103) and non-athletes (*n* = 127). Consequently, two genetic variants were identified: the deletion of one of three adenines in the first intron (c.373+90delA, rs11333758) and a non-synonymous variant in the second exon (c.458A>G, p.Lys(K)153Arg(R), rs1805086). Among all samples, the *MSTN* rs1805086 Lys(K) allele was the most common form in both groups. Homozygous genotype for the less common Arg(R) allele was identified in only one elite canoe rower, and we could find no direct association between rs1805086 and successful results in elite athletes. Surprisingly, the intronic variant (rs11333758) was abundant among all samples. The main finding was that endurance-oriented athletes had 2.1 greater odds of being *MSTN* deletion genotype than non-athletes (13.6% vs. 0.8%). The present study confirms the association of the polymorphism rs11333758 with endurance performance status in Lithuanian elite athletes.

## 1. Introduction

Molecular genetics and exercise genomics now represent the application of a useful tool for the analysis of athletic performance phenotypes, and many debates are taking place about the different roles of biological and environmental contributions to physical performance. Elite athletes appear to emerge as a result of endogenous (such as anatomical, metabolic, functional, or behavioural) characteristics interacting with the exogenous influences of environmental factors (such as training, diet, medical assistance, etc.) [1]. Additionally, gene interactions are relevant for physical performance (such as muscles developing, body composition, training resistance, predisposition to injuries, or behavioural attitude to the competition). Among the many potential genes that have been associated with athletic performance, the myostatin gene (*MSTN*) plays a negative role in muscle development (in proliferation and differentiation) [1,2]. The *MSTN* gene is located on chromosome 2q32.2. This gene encodes myostatin (or growth differentiation factor 8) belonging to the transforming growth factor beta (TGF-β) protein superfamily [1,2,3]. Myostatin was first identified in 1997 as a myokine, which synthesises and releases during muscular contractions [3,4]. Myostatin is synthesized as a preprotein activated by 2 proteolytic cleavages. Removal of the signal sequence is followed by cleavage at a tetrabasic processing site, resulting in a 26-kD amino-terminal propeptide and a 12.5-kD carboxy-terminal peptide, a dimer of which is the biologically active portion of the protein (MIM 601788). It is abundantly expressed in skeletal muscles but is also expressed to a lesser extent in cardiac muscles and fat tissues [2,3,4]. The bioavailability and the function of myostatin dependent of a network of protein interactions. It has been established that myostatin signalling pathway can recruit various intracellular signalling cascades. The myostatin pathway is conserved across diverse species ranging from zebrafish to humans [2]. Mutations in the *MSTN* gene leading to inactive or defective gene product have been shown in skeletal muscle hypertrophy (and/or hyperplasia) phenotypes in mice, cattle, dogs, and sheep [2,5,6]. The first studies using mouse models demonstrated that myostatin is expressed in different muscles throughout the body and during early development and in adults. Knockout mice (Mstn−/−) without functional myostatin are significantly larger than wild-type animals, with 200% more skeletal-muscle mass because of an increase in the size of muscle fibers (hypertrophy) and in the number of myocytes (hyperplasy) [6]. A similar phenotype is seen in some breeds of double-muscled cattle that also have myostatin mutations [6,7]. Furthermore, growing evidence indicates that increased myostatin activity contributes to muscle atrophy, but also regulates the lifespan in mammals [4,8,9]. In addition, myostatin is also involved in the regulation of adipose tissue [10,11]. In knockout mice, adipose tissue size is greatly reduced [10]. The positive effects of myostatin inhibition on adipose tissue seem to be indirect results of changes in muscle mass and muscle metabolism [9,10,11]. Data from scientific literature suggest that myostatin acts in a similar manner in all mammals [7]. The analysis of research conducted on animals has led to greater understanding of the role of myostatin in the human body. A child carrying a mutation in the *MSTN* gene and presenting increased muscle mass and muscle strength was identified, which increased interest in the study of this gene [12]. Functional *MSTN* mutations are very rare in humans. Single nucleotide polymorphisms (SNP) in the *MSTN* gene can have a functional impact on phenotypic traits in individuals [1,11,12,13,14,15,16,17,18,19,20,21,22,23]. These phenotypic consequences include muscle strength [1,13,16], endurance performance [17], left ventricular mass [18], peak bone mineral density [19], sarcopenia [15,20,21], Duchenne muscular dystrophy [22], and obesity [11,23], among others [7,8,14]. Inconsistent results have however been observed with regard to the phenotypic associations of polymorphisms in *MSTN*, which could be explained by the different ethnicity and/or genders of the participants in different studies. Up until now, several polymorphisms (such as rs1805086, rs35781413, rs3791783, rs11681628, and rs7570532) of *MSTN* have been investigated for their associations with muscle hypertrophy or muscle strength of athletes, but the overall findings also have been inconsistent [1,16,17,18,24]. One of the identified SNPs, rs1805086 (A>G replacement in exon 2; p.153K>R (Lys153Arg)), has been associated with a skeletal muscle phenotype such as hypertrophic response in muscles due to strength training [13,16,23,25,26]. To date, research studies on *MSTN* polymorphisms and human muscle phenotypes have provided controversial results, which can be explained, at least partially, by ethnic differences. Apart from this, organisations such as the World Anti-Doping Agency list myostatin inhibitors as a research topic of concern [27].

Given the above, we hypothesised that variants in the *MSTN* gene might be associated with physical performance phenotypes and the status of elite athlete. Thus, we aimed to study the potential role of the *MSTN* gene in the physical performance of athletes by analysing the whole coding sequence of the *MSTN* gene (three exons, including partial flanking intron sequences) in a cohort of Lithuanian elite athletes and in a healthy non-athletes group.

## 2. Materials and Methods

### 2.1. Subjects and Ethics Approval

All procedures in this study conformed with ethical standards concerning scientific research of sport and exercise and were approved by the Lithuanian Bioethics Committee. Written informed consent was obtained from all participants and the study was conducted in compliance with the Declaration of Helsinki.

The study involved 103 Lithuanian elite athletes (67 males and 36 females, participants in and winners of major international competitions, including the European Championships, World Championships and Olympic Games; aged 25.4 ± 6.5 years) and 127 controls (68 male and 59 female healthy, unrelated Lithuanian citizens who were not athletes; aged 32.9 ± 6.7 years). The athletes were stratified into two groups according to the events they compete in: (1) endurance-oriented (marathon runners, long-distance swimmers, and rowers, *n* = 44) and (2) sprint/power-oriented athletes (track and field sprinters (running, throwing, and jumping) and cycling sprinters, *n* = 59). All members of the athlete group and control group were Caucasians.

### 2.2. Phenotypic Data

The phenotypic data of athletes (anthropometric measurements, anaerobic muscle strength (handgrip, counter-movement jump, and stair climbing tests), and maximal oxygen uptake (VO_2_max)) were evaluated. Body height was measured to the nearest 0.01 m with the subject standing with their back to a wall-mounted stadiometer. Weight was measured to the nearest 0.1 kg with calibrated scales. Body mass index (BMI; in kg/m^2^) was calculated as weight (in kg) divided by height (in m^2^). Highly trained athletes may have a high BMI because of increased muscularity rather than increased body fat. Total body fat mass (FM) was determined by measuring the size of the thickest places of the forearm, humeral area, thigh, and calf and by using a calliper to measure the thickness of the skin, thus determining the amount of subcutaneous fat. Anaerobic power was determined with three different tests, using the technique recommended by Brown and Weir [28]. Short-term explosive muscle power (STEMP, vertical jump test, in W) was measured by asking the subject to perform a maximal vertical jump (on a contact platform), and the power output expressed per unit of body weight was measured according to Bosco procedures and modifications by Linthorne [28,29]. Anaerobic alactic muscular power (AAMP, in W) was estimated by a stair-climbing test proposed by Margaria [30]. Maximal isometric force and peak strength of the forearm muscles (handgrip test) was measured by a Jamar hydraulic hand dynamometer and expressed in kilograms. For convenience, we denote left- and right-hand grip strength as LGS and RGS, respectively. Aerobic capacity was determined by using maximum oxygen consumption (VO_2_max). VO_2_max refers to the maximum amount of oxygen that an individual can utilise during maximal or exhaustive exercise. It is measured as millilitres of oxygen used in 1 min per kilogram of body weight (ml/kg/min). The gas exchange was measured using a facemask apparatus attached to a continuous, breath-by-breath monitoring system (*Oxycon Mobile*, Germany). The athletes were evaluated while pedalling on a cycle ergometer (*Ergoselect 100 P*, Germany) or running on a treadmill (*Cosmos Mercury*, Germany) according to an incremental procedure until exhaustion. Oxygen uptake, pulmonary ventilation, ventilatory equivalents for oxygen and carbon dioxide, and end-tidal partial pressure of oxygen and carbon dioxide were measured during each test. All assessments were carried out by trained individuals according to a standardised procedure.

### 2.3. Sequencing Analysis of the MSTN Gene

Genomic DNA was extracted from peripheral blood leukocytes by the standard phenol-chloroform method. All coding sequences of the *MSTN* gene (three exons, including partial flanking intron sequences) were examined by PCR amplification and direct sequencing (using an automatised *Sanger* sequencing technique). The four pairs of primers for the amplification (the three exons of *MSTN* were divided into four fragments using Primer3 software (http://bioinfo.ut.ee/primer3-0.4.0/, accessed on 1 February 2016) and sequencing analysis were designed based on GenBank sequences. Exon 1 was amplified as a 565-bp fragment, including 131 and 373 bp of the 5′ untranslated and protein coding regions, respectively, as well as 61 bp of intron 1, using two primers 1F: 5′-GCTTGGCATTACTCAAAAGCA-3′ and 1R: 5′-TTACATACAGGCCAACAGCTTG-3′. Exon 2 was amplified using primers 2F: 5′-ATGGAGGGGTTTTGTTAATGG-3′ and 2R: 5′-GGCTACCGTTGGGGTAAGAT-3′, which resulted in a 570-bp fragment, including 374 bp of exon 2, 97 bp of intron 1, and 99 bp of intron 2. With the use of four primers (3.1F: 5′-CATTCCTTAATGCTGTGCCTTT-3′, 3.1R: 5′-GTTTGCTTGGTGTACCAGATGA-3′ and 3.2F: 5′-CGATGCTGTCGTTACCCTCT-3′, 3.2R: 5′-TCTAGGCCTATAGCCTGTGGTACT-3′), exon 3 and the 3′ untranslated region were amplified as two fragments, 391 bp and 390 bp, including 61 bp of intron 2, 381 bp of the protein coding region, and 374 bp of the 3′ untranslated region. The amplified products were analysed on a 1.5% agarose gel and visualized by ethidium bromide staining. The sequencing of the *MSTN* gene was performed in four steps: (1) purification of PCR products by the ExoSAP method, (2) cycle sequencing, (3) ethanol/sodium acetate precipitation, (4) nucleotide sequencing by automated capillary electrophoresis using the 3130xl Genetic Analyzer (Applied Biosystems™, Life Technologies, (2012) Foster City, CA, USA) according to the directions of the manufacturer. Sequencing results were compared with the wild-type sequence (Genbank). Sequences were visually inspected and were aligned for each individual, with use of the programs Sequencing Analysis Software 5.2 (Applied Biosystems™, Life Technologies, (2012) Foster City, CA, USA) and Chromas Lite 2.1 (http://technelysium.com.au/wp/chromas/, accessed on 1 September 2017).

### 2.4. Data Analysis

The SPSS statistical package (IBM Corp. Released 2012. IBM SPSS Statistics for Windows, Version 21.0. Armonk, NY: IBM Corp.) and RStudio environment v.1.2.5033 were used to perform all statistical evaluations. A chi-squared test was used to confirm that the genotype frequencies were in Hardy–Weinberg equilibrium and to compare alleles and genotype frequencies between athletes and controls, and between athletes from different sports. The average differences for each genotype of the phenotypic indexes of Lithuanian athletes were evaluated by using a Student’s *t*-test or one-way dispersion analysis (ANOVA). All data were presented as mean ± standard deviation (SD). Differences between groups were considered as statistically significant when *p*-value was below 0.05.

## 3. Results

All three coding regions (including partial flanking intron sequences) of the *MSTN* gene were successfully PCR amplified from the genomic DNA of the subjects. Thus, 230 DNA samples (103 athletes and 127 controls) were subjected to direct sequencing. Sequencing results of the exon 1 encompassing region disclosed completely normal sequences in all of the samples. We did however frequently find a point mutation in intron 1 of the *MSTN* gene, a deletion of one of three adenines (AAA→AA) at position 88–90 bp (rs11333758, NM_005259.2:c.373+90delA). To our knowledge, there has been only one report so far on the functional relevance of this polymorphism [18]. Hereafter the genotypes of this SNP in *MSTN* will be referred to as homozygous for the wild type (AA), heterozygous for the deletion (A/–), and homozygous for the deletion (−/−). In the exon 2 encompassing region, overlapping A and G peaks at the 458th nucleotide of the *MSTN* cDNA (NM_005259.2:c.458A>G) were uncovered in one sample. *MSTN* c.458A>G corresponded to the known polymorphism rs1805086 that changes a Lys (K) to Arg (R) at the position corresponding to the 153rd amino-acid residue of myostatin (NP_005250.1:p.Lys153Arg) [7,9,13,14,15,16,17,22,23,24,26]. Besides, examination of the exon 3 encompassing region did not disclose any nucleotide changes in the genomic samples.

The SNP in the second exon of *MSTN* (rs1805086, K153R) and the homozygous genotype KK (and K allele) was the most common form in both elite athletes and the control group. Moreover, among all investigated samples, homozygous genotype RR (and rare R allele) was found in only one elite canoe rower (a participant in international competitions, including the World and European Championships) and was not identified in the control group. Based on this evidence, we performed a distinct phenotypical data analysis of the athlete with the *MSTN* RR genotype: the athlete was 180-cm tall, weighed 86.5 kg, and had a high value of muscle mass (46–49 kg) and relatively low-fat mass (6–8 kg). His body mass index was calculated to be 26.0–27.0 kg/m^2^, lean body mass was 64.0 kg, fat-free mass index was 24.8 kg/m^2^, handgrip strength was 62 kg, short-term explosive muscle power was 2400–2700 W (measured by vertical jump test), anaerobic alactic muscular power was 1600–1900 W (measured by stair climbing test), maximum oxygen consumption was 70 mL/min/kg, and lung volume was 6.7 L. During the annual sporting season, the athlete’s phenotypical data remained at the same levels.

The intronic variant (rs11333758, c.373+90delA) of the *MSTN* gene was abundant in all samples. The results of the distribution of this variant (rs11333758) in Lithuanian elite athletes versus controls are presented in Table 1. In both athletes and the control group, genotype distributions were in agreement with the Hardy–Weinberg equilibrium (in all groups tested separately, *p* > 0.05). There was a significantly higher frequency of the homozygous deletion genotype (−/−) in the athlete group compared with the control group (9.7% vs. 0.8%, odds ratio (OR) (95% CI) = 1.8 (1.1–2.8), *p* = 0.006). The frequency of deletion allele (24.8%) in all athletes and in the endurance-oriented group (28.4%) was significantly higher than in the control group (15.7%, *p* < 0.05). Additionally, the deletion genotype was more frequent in endurance-oriented athletes than in non-athletes (13.6% vs. 0.8%, OR (95% CI) = 2.1 (1.2–3.8), *p* = 0.001).

Significant differences were also obtained when genotype frequencies were compared with respect to gender: between the male athletes and male controls (AA 64.2%; A/– 26.8%; (−/−) 9.0% vs. AA 64.7%; A/– 35.3%; (−/−) 0%; *p* = 0.032) and between the female athletes and female controls (AA 52.8%; A/– 36.1%; (−/−) 11.1% vs. AA 74.6%; A/– 23.7%; (−/−) 1.7%; *p* = 0.037). The frequency of the deletion allele (29.0%) in the female athlete group was significantly higher, especially in endurance-orientated female athletes (38.2%), than in the female control group (14.0%, *p* < 0.05). There was a significantly higher frequency of the homozygous deletion genotype (−/−) in endurance-orientated female athletes (AA 47.1%; A/– 29.4%; (−/−) 23.5%) and in sprint/power-orientated male athletes (AA 65%, A/– 25%; (−/−) 10%) than in the controls with respect to gender (*p* < 0.05).

The phenotypical variables of the control group were not measured due to various limitations. Phenotypic values in each group of Lithuanian elite athletes were specific to the sport category (Table 2). In additional, male and female athletes were analysed separately given the known gender-specific influences on phenotypic measures. We determined that for all variables analysed the mean values were significantly different with respect to gender and sports groups (*p* < 0.01, except height and fat mass). However, the mean differences between genders within sports groups showed less pronounced results. The average value of anaerobic power variables, such as STEMP and left and right hand grip strength, was significantly higher for males in all groups. Furthermore, the phenotypic characteristics of the study participants were categorised according to their *MSTN* rs11333758 (c.373+90delA) polymorphic genotypes. It is notable that we did not determine statistically significant differences between the *MSTN* rs11333758 genotypes and physical performance phenotypes of whole group of athletes (*p* > 0.05). Irrespective of genotype, all the sprint/power-oriented athletes had higher muscle mass and handgrip strength compared with the endurance group. The anaerobic power variable STEMP and hand grip strength were significantly higher for *MSTN* A/A homozygous and A/**–** heterozygous of the sprint/power group compared with the endurance group (*p* < 0.05).

## 4. Discussion

Myostatin is a negative regulator of muscle growth that is attracting attention as a candidate gene for physical performance traits. Myostatin might exert its effect through its influence on skeletal muscles (as well as adipose tissue) that in turn control human physical activity, aging and lifespan [1,8,9,11,14,15,21,23,25,31]. It has been established that both aerobic exercise and resistance training in humans attenuate myostatin expression and myostatin inactivation seems to potentiate the beneficial effects of endurance exercise on metabolism [4]. To date, however, no study has been undertaken to investigate the association between *MSTN* variation and physical performance phenotypes of Lithuanian elite athletes. Hence, the present study was designed to establish the presence or absence of such an association in order to address the paucity of this information. The *MSTN* gene was chosen based on the aforementioned functional studies to concisely obtain data regarding athletic performance and exercise genetics. In particular, the *MSTN* gene of athletes who have competed successfully in World and European championships and the Olympic Games was analysed. *MSTN* was investigated at the molecular level and functional (phenotypical) association was assessed. Thus, we conducted extensive sequence analysis of the *MSTN* gene in 102 Lithuanian elite athletes and 127 controls. As a result, two genetic variants were identified: (1) the deletion of one of three adenines (AAA→AA) at position 88–90 bp in the first intron (c.373+90delA, rs11333758) and (2) a non-synonymous coding variant in the second exon (c.458A>G, p.Lys(K)153Arg(R), rs1805086).

The main, novel finding of our study was that the polymorphism in intron 1 (rs11333758) of *MSTN* was significantly associated with endurance performance status in Lithuanian elite athletes. In addition, the known *MSTN* rs1805086 polymorphism (homozygous genotype with rare R allele) was identified in only one elite canoe rower. During the annual sporting season, the athlete’s phenotypical data remained at the same levels, which leads to the conclusion that the *MSTN* rs1805086 polymorphism, in combination with other genetic, epigenetic, and environmental factors, may be associated with an athlete’s general physical performance and high results in sports. Thus, in this paper we demonstrated no direct association between the *MSTN* variant in exon 2 (rs1805086) and successful results in elite athletes.

The most common SNP in *MSTN* gene rs1805086 has been previously analysed in various cohorts of different ethnic or geographic origin that showed a higher prevalence of the *MSTN* wild type K allele [14,15,16,21,23,26,31,32]. The average minor allele frequency (MAF) of the rs1805086 polymorphism (R allele) among Caucasians is 3–4%, and the frequency of homozygotes (RR) is about 1% [24]. Interestingly, the polymorphism rs1805086 appears at a relatively high frequency among Africans (22%) but at much lower frequencies among Americans (5%), Europeans (3%; additionally, 0% in Finland) and Asians (0%) (1000 Genomes Project Phase 3) (http://www.ensembl.org/, accessed on February 2021). Data contained in publications showed that the MAF of this SNP is observed at the following rates in the Italians (0.44%) [21], Spanish (2.83%) [13], Belgians (0.87%) [26], Japanese (0.00%) [16,22,31], and Indians (15.82%) [23]. Saunders et al. (2006) provided compelling evidence that *MSTN* rs1805086 has been subject to recent positive selection, suggesting that it is associated with functional differences [7]. Recent studies demonstrated an association between the *MSTN* rs1805086 polymorphism with lower muscle strength, higher obesity risk, and extreme longevity, but the molecular basis of these associations has not been clarified [8,9,11,14,15,21,23,25,31,32]. Santiago et al. (2011) found that the *MSTN* rs1805086 polymorphism (KR genotype) affects the ability to produce peak power during muscle contractions in non-athletic young men [13]. Khanal et al. (2020) identified *MSTN* rs1805086 KK homozygotes as the favourable genotype for thicker biceps brachii in the elderly women [33]. The RR genotype of this SNP was found to be associated with a high risk of obesity in Asian Indians [23]. The R allele was also shown to be associated with longevity in Spanish and Italian centenarians [9,14]. Li et al. (2014) demonstrated the association between the rs1805086 polymorphism and strength training-induced muscle hypertrophy among Han Chinese men [16]. This is consistent with the finding of Kostek et al. (2009), who showed that the *MSTN* rs1805086 polymorphism was associated with muscle size among African Americans [25]. However, the same authors found no association between this SNP and muscle size among participants from other ethnicities [25]. Similarly, the lack of association of the rs1805086 polymorphism was observed by Thomis et al. (2004) among Belgians [26]. Moreover, none of the common variants in *MSTN* appeared to influence the ability to attain top-level endurance performance in the Genathlete study [17]. Up to date, published data on the *MSTN* rs1805086 polymorphism and human muscle phenotypes (at baseline or in response to training) have yielded controversial results, at least in adults of young or medium age. Inter-ethnic differences in *MSTN* rs1805086 allele frequencies, gender related differences, and the low allelic frequency of the R allele (limiting the possibility of studying large groups of people carrying the R variant) are important reasons for this controversy [13].

In the present study, we showed that the intronic variant rs11333758 (c.373+90delA, a deletion of one nucleotide A) was abundant among elite athletes and non-athlete controls. Despite being located in the intron and without altering the amino acid composition of myostatin, this variant purportedly affects *MSTN* gene expression and myostatin function. This SNP was found to have a MAF of >1.0% in our population, and the genotype frequencies of this SNP did not deviate from the Hardy–Weinberg equilibrium (*p* > 0.05). The MAF of rs11333758 in this study (15.7% in non-athlete controls and 24.8% in athletes) was similar to findings in the European population (22%) from the Ensemble database (1000 Genomes Project Phase 3). The frequency of the minor (–) allele (28.4%) and the homozygous for deletion (−/−) genotype (13.6%) in the endurance group was significantly higher than in the control group (*p* < 0.05) (Table 1). This confirms our primary hypothesis that the SNP in the *MSTN* gene is associated with athlete status, especially in endurance sports. Interestingly, it has already been shown that myostatin is expressed at higher levels in slow-twitch muscle fibers and may therefore have a more significant functional impact in this muscle group [34]. To our knowledge, there has been only one report so far on the functional relevance of the rs11333758 polymorphism in the *MSTN* gene [18]. Karlowatz et al. (2011) analysed the association of an endurance athlete’s heart (left ventricular mass, LVM) with genetic polymorphisms in the insulin-like growth factor 1 (IGF1) signalling pathway including the *MSTN* gene. In this cross-sectional study, the authors conducted an analysis of the entire coding sequence of *MSTN* in a group of 110 elite endurance athletes and 27 male controls. The authors found only one SNP (rs11333758, AAA→AA) in intron 1 of the *MSTN* gene. In the group of 75 male athletes (but not females (*n* = 35) or male controls), carriers heterozygous for genotype (A/–) and homozygous for deletion (−/−) had a significantly lower LVM than carriers homozygous for wild type (A/A). The results of the Karlowatz et al. (2011) study indicate an increased *MSTN* effect for the deletion (–) allele. This means that carriers of at least one deletion allele may show an attenuated training-induced growth response of the heart, resulting in a lesser LVM increase. It is unclear whether the rs11333758 SNP (although untranscribed) is a functional variant or a marker influencing LVM or physical performance phenotype. In any case, it should be noted that this *MSTN* SNP is located in an evolutionarily highly conserved sequence [18].

In addition, the present study demonstrated that significant differences were obtained when genotype frequencies were compared with respect to gender. The frequency of the deletion allele (29.0%) in the group of female athletes was significantly higher, especially in endurance-orientated female athletes (38.2%), than it was in the group of female controls (14.0%) (*p* < 0.05). There was a significantly higher frequency of the homozygous deletion genotype (−/−) in endurance-orientated female athletes (23.5%) and in sprint/power-orientated male athletes (10%) compared with the controls with respect to gender (1.7% female controls and 0% male controls, *p* < 0.05). Interestingly, a gender difference was identified in the effect of rs11333758 SNP on the physical performance of athletes. Our findings demonstrate the association of rs11333758 (deletion allele and genotype) with success in endurance sports, especially in female athletes and additionally in sprint/power-orientated male athletes, and provide evidence for the role for the *MSTN* rs11333758 variant in elite athletic performance. Additionally, we investigated phenotypes of athletes that are related to physical performance to begin creating a chain of evidence linking the polymorphisms of the *MSTN* gene to success in elite sports. Aerobic capacity was determined using maximum oxygen uptake, which is widely accepted as the single best measure of cardiovascular fitness and maximal aerobic power. Instantaneous or explosive power in the lower extremities was measured by a vertical jump test and stair-climbing test. Maximal isometric power of the forearm muscles was measured using handgrip strength. Among the performance tests used for athletes, strength and anaerobic power tests are the most indicative of muscle properties. In fact, no statistical significance emerged between the rs11333758 genotype and phenotype values of elite athletes.

It is important that our study was the first to demonstrate the association of the rs11333758 SNP with the status of elite athlete. Further studies are needed to replicate those findings and to confirm whether this SNP is associated with performance in endurance-oriented sports.

We acknowledge the limitations of the present study. First, we studied a one candidate gene and recognise that genetic association studies represent only the first steps toward understanding the genetic factors influencing physical performance traits. However, candidate gene association studies are crucial to the direction of modern genetic analyses of physical capacity. Second, participant nationality was limited to Lithuania, meaning the associations described in the present study cannot be generalised to athletes from other countries. It has already been shown that many genetic variants that have a significant association with physical performance in several studies of one population may not necessarily have the same association in another. We also recognise that the sample size is small by the standards of modern genomic studies. Regardless of the relatively small number of participants, our study groups demonstrate unique and clearly distinctive phenotypes. Each group contains elite-level athletes from a well-defined sports category that has known prime determinants for success. Although in such small samples true positive associations could be masked, we believe that this paper gives valuable insights about the physical capacity of elite athletes and the *MSTN* gene. Further research is needed to determine the possible association between *MSTN* polymorphisms (rs1805086 and rs11333758) and the physical performance phenotype in elite athletes.

## 5. Conclusions

The present study confirms associations of the *MSTN* intronic variant rs11333758 with the status of elite athlete, indicating that this SNP is likely one of several variants affecting the development of physical performance phenotypes. It is notable that our data for the first time suggest a strong role of *MSTN* rs11333758 (deletion allele) in determining endurance sports success and of its association with Lithuanian elite athletes’ status. Additional studies of elite athletes from different nationalities or ethnic groups are required to further substantiate our findings. This study also demonstrated that the common polymorphism rs1805086 in the second exon of the *MSTN* was found in only one high-level canoe rower. As there are substantial differences in the allele frequencies of this SNP rs1805086 among different ethnic populations, it is quite possible that this variant contributes to physical performance in athletes. From a functional point of view, the *MSTN* gene appears to be feasible candidate for physical performance. However, we found no significant association between *MSTN* genotypes and particular functional capacity phenotype.

## Figures and Tables

**Table 1 genes-12-00757-t001:** Genotype and allele distributions of the *MSTN* rs11333758 polymorphism in Lithuanian elite athletes and controls.

Groups	N	Allele Frequency, %		*p*-Value Compared with Control	*MSTN* c.373+90delA Genotype Frequency, *n* (%)			*p*-Value Compared with Control
		A	–		AA	A/–	(−/−)	
Endurance-oriented	44	71.6	28.4	0.014	25 (56.8)	13 (29.6)	6 (13.6)	0.001
Sprint/power-oriented	59	78.0	22.0	0.140	37 (62.7)	18 (30.5)	4 (6.8)	0.060
All athletes	103	75.2	24.8	0.022	62 (60.2)	31 (30.1)	10 (9.7)	0.006
Controls	127	84.3	15.7	-	88 (69.3)	38 (29.9)	1 (0.8)	-

**Table 2 genes-12-00757-t002:** Phenotypic characteristics of Lithuanian elite athlete groups.

Phenotypic Characteristics	Endurance-Oriented (*n* = 44)	Sprint/Power-Oriented (*n* = 59)	*p*-Value
Height, cm	180.2 ± 9.1	179.8 ± 8.7	0.597
Weight, kg	72.5 ± 10.7 *	77.8 ± 12.6 *	0.007
BMI, kg/m^2^	22.2 ± 1.6 *	23.8 ± 3 *	0.000
Fat mass, kg	8 ± 2.2	7.8 ± 3.8	0.637
Muscle mass, kg	39.6 ± 8.9 *	43.1 ± 8.7 *	0.015
Right handgrip strength, kg	54.1 ± 8.3 *	61.4 ± 6.1 *	0.000
Left handgrip strength, kg	52.7 ± 7.1 *	58.6 ± 10.9 *	0.000
STEMP, W	1858.5 ± 315.63 *	2366.3 ± 372.1 *	0.000
AAMP, W	1131 ± 154.3 *	1298.5 ± 208.8 *	0.000
VO_2_max, mL/min/kg	70.6 ± 3.3 *	64.6 ± 4.7 *	0.000

Data presented as means (± standard deviation). BMI—body mass index, STEMP—short-term explosive muscle power, AAMP—anaerobic alactic muscle power; VO_2_max—maximum oxygen uptake.

## Data Availability

The data presented in this study are available on request from the corresponding authors.

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
