# Peer review of "Variants in the Myostatin Gene and Physical Performance Phenotype of Elite Athletes"

_genes, 2021, doi:10.3390/genes12050757_

Round 1
Reviewer 1 Report
Manuscript under this review concerns very interesting concept of interaction between the MSTN gene and physical performance by analysing the sequence of the MSTN gene in Lithuanian elite athletes. This is a new and original contribution, but also gives support other findings in this area.
Manuscript reports results of well-designed study. The purpose of this paper is relevant and important. Title mostly identifies theme of the paper. Abstract is sufficiently informative. In the introduction section, paper has adequate theoretical reflection on subject matter. The methodology is appropriate and well described, the data are correctly analyzed, clearly presented and support the conclusions. Discussion relates findings of this paper to existing knowledge on an international level. The authors highlight the aims and significance of their work as well as the potential limitations. Interpretations and conclusions sound justified by the data.
The experiment presented in the paper is well planned and conducted. The authors have made a good effort to address the issues raised and I have no major critical remarks.
I do identify a need of only minor corrections:
- Authors should check the text and improve the punctuation mistakes, e.g. Key words: after “Sprint and Power” it should be”;” not “,”; it should be 5% not 5 %. Whether or not you put a comma before and depends on how you’re using and. Needs correction throughout the entire manuscript.
- Line 33-37: the sentence is too long
- Introduction: Please add information on the structure of the myostatin protein and its mechanism of its biological action.
- Discussion: The authors could try to focus a bit more on the potential biological mechanisms behind the potential associations.
Author Response
Response 1: We have responded more comprehensively to Reviewer’s comments regarding our correction (punctuation mistakes) throughout the entire manuscript.
Response 2: We made the sentence shorter:
Among the many potential genes that have been associated with athletic performance, the myostatin gene (MSTN) plays a negative role in muscle development (in proliferation and differentiation).
Response 3: Thank you for raising this point. In response to Reviewer’s requests, we have included comprehensive information surrounding the structure of the myostatin protein and its mechanism of its biological action.
We have added the following (inserted information is in red) to Introduction section of the manuscript. Line 33-47:
Among the many potential genes that have been associated with athletic performance, the myostatin gene (MSTN) plays a negative role in muscle development (in proliferation and differentiation) [1,2]. The MSTN gene is located on chromosome 2q32.2. This gene encodes myostatin (or growth differentiation factor 8) belonging to the transforming growth factor beta (TGF-ß) protein superfamily [1–3]. Myostatin was first identified in 1997 as a myokine, which synthesises and releases during muscular contractions [3,4]. Myostatin is synthesized as a preprotein activated by 2 proteolytic cleavages. Removal of the signal sequence is followed by cleavage at a tetrabasic processing site, resulting in a 26-kD amino-terminal propeptide and a 12.5-kD carboxy-terminal peptide, a dimer of which is the biologically active portion of the protein [MIM 601788]. It is abundantly expressed in skeletal muscles but is also expressed to a lesser extent in cardiac muscles and fat tissues [2–4]. The bioavailability and the function of myostatin dependent of a network of protein interactions. It has been established that myostatin signalling pathway can recruit various intracellular signalling cascades. The myostatin pathway is conserved across diverse species ranging from zebrafish to humans [2].
Response 4: We thank Reviewer for raising this point. The addition of this information certainly helps to clarify potential associations. We have added the following sentence to Discussion section:
Line 255-257: It has been established that both aerobic exercise and resistance training in humans attenuate myostatin expression and myostatin inactivation seems to potentiate the beneficial effects of endurance exercise on metabolism [4].
Line 320-322: Despite being located in the intron and without altering the amino acid composition of myostatin, this variant purportedly affects MSTN gene expression and myostatin function.
We are grateful to Reviewer #1 for raising all of these important points, and we feel that the changes we have made as a consequence have improved the quality of the manuscript.
Reviewer 2 Report
The manuscript by Gineviciene et al. entitled "Variants in the myostatin gene..." describes the occurrence of polymorphisms in the myostatin (MSTN) gene and the association with physical performance in Lithuanian athletes. The key finding was that endurance athletes had the highest incidence of the intronic deletion SNP and the exonic non-synchronous SNP showed no direct association.
This manuscript is interesting and well-written. There are a few minor questions.
On line 153, consider rephrasing "were inspected by eye" to "visually inspected."
In the discussion section on lines 244-246, are ten references necessary to support this statement. The same question is posed for lines 282-285. If so, then there are no objections.
Author Response
We are grateful to Reviewer for raising of these points.
Point 1: On line 153, consider rephrasing "were inspected by eye" to "visually inspected."
Response 1: We rephrasing "were inspected by eye" to "visually inspected."
Point 2: In the discussion section on lines 244-246, are ten references necessary to support this statement. The same question is posed for lines 282-285. If so, then there are no objections.
Response 2: All references are necessary to support this statement.